# A Transformer-Optimized Deep Learning Network for Road Damage Detection and Tracking

**DOI:** 10.3390/s23177395

**Published:** 2023-08-24

**Authors:** Niannian Wang, Lihang Shang, Xiaotian Song

**Affiliations:** 1School of Water Conservancy and Transportation, Zhengzhou University, Zhengzhou 450001, China; 2School of Engineering and Technology, China University of Geosciences (Beijing), Beijing 100083, China

**Keywords:** road damage detection, object tracking, self-attention mechanism, transformer

## Abstract

To solve the problems of low accuracy and false counts of existing models in road damage object detection and tracking, in this paper, we propose Road-TransTrack, a tracking model based on transformer optimization. First, using the classification network based on YOLOv5, the collected road damage images are classified into two categories, potholes and cracks, and made into a road damage dataset. Then, the proposed tracking model is improved with a transformer and a self-attention mechanism. Finally, the trained model is used to detect actual road videos to verify its effectiveness. The proposed tracking network shows a good detection performance with an accuracy of 91.60% and 98.59% for road cracks and potholes, respectively, and an F1 score of 0.9417 and 0.9847. The experimental results show that Road-TransTrack outperforms current conventional convolutional neural networks in terms of the detection accuracy and counting accuracy in road damage object detection and tracking tasks.

## 1. Introduction

For economic development and social benefits, the health of roads is crucial. In daily life, repeated crushing by vehicles can cause damage to the structural layer of the road, which in turn produces cracks, potholes and other damage. The road performance and load carrying capacity will suffer as a result of pavement degradation [1,2]. If pavement damage is not repaired in a timely manner, rain and snow, as well as vehicle loads, will deepen the degree of pavement damage, which will seriously affect people’s travel and safety and thus have an impact on social benefits. Therefore, regular maintenance of roads is very important. For road maintenance, one of the main aspects lies in efficient and accurate road damage detection. Currently, manual inspection and analysis is the main method of detecting pavement damage in China; however, manual inspection is often tedious and inefficient [3]. Although manual inspection has obvious operational advantages, when the inspector is inexperienced, the assessment of the degree of damage can be inaccurate, thus adversely affecting the pavement evaluation process [4,5,6]. The drawbacks of these manual inspections mean that this method no longer meets the increasing requirements of modern society for road damage detection.

### 1.1. Related Works

#### 1.1.1. Conventional Methods

In addition to the above manual detection methods, conventional methods of road damage detection include automatic detection and image processing techniques. With the development of research and technological support, the usage of automatic road damage detection is constantly expanding, with conventional equipment such as infrared or sensor-equipped road inspection vehicles [7,8]. However, due to the complexity of the actual environment in the road detection process, automated detection equipment is often unable to meet the actual needs in terms of recognition accuracy and speed, and this type of equipment often incurs higher hardware costs, corresponding to an increase in detection costs. For example, some vibration-based detection methods are suitable for real-time assessment of pavement conditions [9], but they cannot measure pavement damage in areas outside the vehicle wheel path or identify the size of pavement damage. Laser-measurement-based inspection methods use special equipment, such as a laser scanner, mounted on a separate inspection vehicle [10,11,12,13] to convert the pavement into a three-dimensional object in a coordinate system, and this method allows for the direct calculation of various metrics for an accurate evaluation of pavement condition. However, real-time processing at high speeds is difficult and relatively expensive due to the increased amount of computation required. Compared with the high cost of automatic detection, the benefits of image processing technology include a great effectiveness and low cost. As technology advances, its recognition accuracy also gradually improves. Therefore, numerous researchers have chosen to use image processing methods for the detection of pavement damage [14,15,16]. Traditional image processing methods use manually chosen features, such as color, texture and geometric features, to first segment pavement faults, and then machine learning algorithms are used to classify and match them for pavement damage detection purposes. For instance, Fernaldez et al. [17] began by preprocessing cracked photos of a road in order to highlight the major aspects of the cracks, and then chose a decision tree heuristic algorithm and finally achieved classification of the images. Rong G et al. [18] performed entropy and image dynamic threshold segmentation of pavement crack pixels based on thresholds obtained from image histograms as a way to classify cracked and non-cracked pixels. Bitelli G et al. [19] proposed another application of image processing to crack recognition, focusing and obtaining additional noise-free images of specific cracks. Li Q et al. [20] presented an image processing algorithm for accuracy and efficiency, which was specifically used for fast evaluation of pavement surface cracks. Song E P et al. [21] proposed an innovative optimized two-phase calculation method for primary surface profiles to detect pavement crack damage. Traditional image processing techniques cannot, however, meet the requirements of model generalization capability and resilience in real-world engineering through manually planned feature extraction due to the complexity of the road environment. For example, it is often impossible to segment an image effectively when it contains conditions such as uneven illumination.

#### 1.1.2. Deep Learning Methods

The issues with the aforementioned conventional methods can be successfully resolved thanks to the recent rapid advancements in artificial intelligence and deep learning technology. Deep learning has advantages over the aforementioned techniques, including the absence of manual feature extraction and good noise robustness. With their strong feature extraction capabilities, deep-learning-based models are used more and more, for example, convolutional neural networks [22] are commonly employed in image classification [23], object detection [24] and semantic segmentation [25]. Nu et al. [26] proposed a unique method for detecting tunneling defects based on a masked region convolutional neural network (RCNN) and optimized the network with a path-enhanced feature pyramid network (PAFPN) and an edge detection branch to increase the detection accuracy. Wang et al. [27] used an improved network model based on Faster RCNN to detect and classify damaged roads, and used data augmentation techniques before training to address the imbalance in the number of different damage datasets to achieve better network training results. In the same vein, for crack detection, Kaige Zhang et al. [28] suggested a depth-generating adversarial network (GAN), which successfully addressed the issue of data imbalance, thus achieving a better training effect and a higher detection accuracy. Yi-zhou Lin et al. [29] suggested a cross-domain structural damage detection method based on transfer learning, which enhanced the performance of damage identification. Wang Zifeng et al. [30] used DeepLabV3 + model to achieve precise segmentation of certain building site objects and three-dimensional object reconstruction. With the help of fully convolutional networks (FCN), Yang et al. [31] were able to successfully identify cracks at the pixel level in pavement and wall images, but there was still a shortcoming of poor detection of small cracks. Jeong et al. [32] improved a model based on You Only Look Once (YOLO)v5x with Test-Time Augmentation (TTA), which could generate new images for data enhancement then combine the original photographs with the improved images in the trained u-YOLO. Although this method achieved a high detection accuracy, the detection speed was not good. Many other researchers have worked to test lightweight models. Shim et al. [33] developed a semantic segmentation network with a small volume. They improved the network’s parameters, but at the same time affected the detection speed of the model. Sheta et al. [34] developed a lightweight convolutional neural network model, which had a good crack detection effect. However, this model still had the problem of a single application scenario and could not deal with multiple road damage detection. Guo et al. [35] improved the model based on YOLOv5s to achieve the purpose of detecting a variety of road damage, and achieved a high accuracy in damage detection. However, the improved model was somewhat higher in weight, and meeting the criteria of embedded devices proved difficult. In addition, Ma D. et al. [36] proposed an algorithm called YOLO-MF that combines an acceleration algorithm and median flow for intelligent recognition of pavement cracks, achieving high recognition accuracy and a good PR curve. All of the above researchers have made reasonable contributions to road damage detection, but there are some deficiencies. For example, the models only detect crack damage, they cannot find a reasonable balance between detection efficiency and accuracy, they cannot effectively detect damage in road videos, etc. These are problems that still need to be studied and solved.

YOLOv5 is a single-stage target detection algorithm. Four versions of the YOLOv5 single-stage target detection model exist: YOLOv5s, YOLOv5m, YOLOv5l and YOLOv5x. For this study, the fastest and smallest model, YOLOv5s, with parameters of 7.0 M and weights of 13.7 M, was selected. YOLOv5 makes the following improvements compared to YOLOv4: For input side, the model training phase makes use of mosaic data augmentation, adaptive anchor frame computation and adaptive picture scaling. The benchmark network makes use of the FOCUS structure and the Cross Stage Partial (CSP) structure. In the Neck network, between the Backbone and the final Head output layer, the Feature Pyramid Network (FPN)_Path Aggregation Network (PAN) structure is added. The loss function named Generalized Intersection over Union Loss (GIOU_Loss) is added to the Head output layer during training and predicts the Distance-IOU_nns of the screening frame.

As shown in Figure 1, the algorithm framework is split into three major sections: the backbone network (Backbone), the bottleneck network (Neck) and the detection layer (Output). The Backbone consists of a focus module (focus), a standard convolution module (Conv), a C3 module and a spatial pyramid pooling module (SPP). In YOLOv5, the network architecture is the same for all four versions, and two variables determine the network structure’s size: depth_multiple and width_multiple. For instance, the C3 operation of YOLOv5s is performed just once, while YOLOv5l is three times as deep as v5s and three C3 surgeries will therefore be carried out. Since the one-stage network YOLOv5s technique leverages multilayer feature map prediction, it produces improved outcomes in terms of detecting speed and accuracy.

#### 1.1.3. Aircraft-Based Evaluation Methods

Manual and automated detection methods are ground-based evaluation methods. Another method of assessing the pavement surface is through aerial observation. Aircraft-based evaluation methods are more efficient, cost-effective and safer than labor-intensive evaluation methods. Su Zhang et al. [37] explored the utility of the aerial triangulation (AT) technique and HSR-AP acquired from a low-altitude and low-cost small-unmanned aircraft system (S-UAS), and the results revealed that S-UAS-based hyper-spatial resolution imaging and AT techniques can provide detailed and reliable primary observations suitable for characterizing detailed pavement surface distress conditions. Susan M. Bogus et al. [38] evaluated the potential of using HSR multispectral digital aerial photographs to estimate overall pavement deterioration using principal component analysis and linear least squares regression models. The images obtained from aerial photography can also be used to train models for pavement damage recognition. Ahmet Bahaddin Ersoz et al. [39] processed a UAV-based pavement crack recognition system by processing UAV-based images for support vector machine (SVM) model training. Ammar Alzarrad et al. [40] demonstrated the effectiveness of combining AI and UAVs by combining high-resolution imagery with deep learning to detect disease on roofs. Long Ngo Hoang, T et al. [41] presented a methodology based on the mask regions with a convolutional neural network model, which was coupled with the new object detection framework Detectron2 to train a model that utilizes roadway imagery acquired from an unmanned aerial system (UAS).

### 1.2. Contribution

Aiming at the problem of repeated missed detections due to a low detection accuracy during video detection of pavement damage, the paper’s primary contribution is to propose and train a tracking and counting model named Road-TransTrack and improve the tracking model by using a transformer and a self-attention mechanism, which increases the detection precision of pavement damage when the model is tracking and achieves accurate counting of damage without damaging the detection speed, making it more appropriate for work detecting pavement damage.

## 2. Methodology

### 2.1. Transformer and Self-Attention

The transformer model is an attention-based neural network architecture that learns interdependencies between sequences through the self-attention mechanism. In a one-dimensional signal classification task, the signal can be considered as a sequence, and the transformer model can be employed to study the interdependence of various points in the sequence. Then, the signal is classified based on the learned information. As shown in Figure 2, based on the correlation between the input samples, a self-attention network is built. Initially, the input sequence x, as shown in Formulas (1)–(3), is multiplied by the weight matrices (*W_k_*, *W_v_*, *W_q_*) to obtain the key vector *k_i_*, the value vector *v_i_* and the query vector *q_i_*, respectively.
(1)$$k_{i}=W_{k}x_{i}$$
(2)$$v_{i}=W_{v}x_{i}$$
(3)$$q_{i}=W_{q}x_{i}$$

Secondly, the key vector is multiplied by the query vector, as shown in Formula (4), and the weight vector a_i_ can be obtained under the softmax function processing. The weight vector represents the correlation of xi with the sequence of [*x*_1_, *x*_2_, …, *x_n_*] of the sequence, i.e., the degree of attention of *x_i_*.
(4)$$a_{i}=Softmax\left(\left[k_{1}^{T},k_{2}^{T}{,k}_{3}^{T},\cdots ,k_{n}^{T}\right]q_{i}\right)$$

After that, as demonstrated by Formula (5), the product of the value vector and the weight vector is the semantic vector *c_i_*, where the value vector represents the value of each input *x_i_*.
(5)$$c_{i}=\left[v_{1}^{T},v_{21}^{T},v_{3}^{T},\cdots ,v_{n}^{T}\right]a_{i}$$

In the end, the distribution of probabilities can be obtained by softtmax function processing and the corresponding output can be obtained by label coding.

The transformer is a pile of self-attention networks, which, in contrast to typical models, uses only self-attention mechanisms as a way to reduce computational effort and not corrupt the final experimental results [42]. As shown in Figure 3, the transformer model has two main parts: an encoder and a decoder. The input patches are fed into the multi-headed self-attention network, which is a type of self-attention network. The multi-headed self-attention network divides the result into eight subspaces and more relevant information can be learned in different subspaces [43]. To improve the deep network, residual connectivity and layer normalization are employed in the full network. As demonstrated by Formula (6), the multilayer perceptron (MLP) consists of two fully connected layers and a nonlinear activation function.
(6)$$MLP(x)=max(0,xW_{1}+b_{1})W_{2}+b_{2}$$

### 2.2. Road-TransTrack Detection and Tracking Model

Traditional deep-learning-based pavement damage detection algorithms are often effective in obtaining the class and location of damage. However, for sequences of consecutive frames, conventional detection algorithms cannot effectively identify the same impairment and cannot accurately count multiple impairments. In this study, the proposed detection tracking model called Road-TransTrack can solve the above problem. Detection is a static task that generally finds regions of interest based on a priori knowledge or salient features. Tracking, however, is a fluid job, finding the same thing in a series of successive frames by means of characteristics carried over from the earlier frame. The tracking task checks the picture similarity of the previous and current frames to find the best matching position to find the target’s dynamic path.

As illustrated in Figure 4, successive frames of the pavement video are first fed into the model, defects are detected when they first appear in frame Ft and the amount of defects is increased by one. The frames F_t_ and F_t+1_ are then fed into the tracking model. This damage continues to be tracked till it vanishes from the video, and IOU (Intersection over Union) matching is performed between the tracked and detected frames to obtain the tracking result. The detection and counting of the next damage continue. Finally, the overall number of discovered defects is determined. Meanwhile, the network is improved with the transformer to enhance the performance of the network.

## 3. Dataset Construction

### 3.1. Data Collection

Like the deep convolutional neural network model, the transformer-improved network model also necessitates a lot of image data as the dataset. Images in today’s road damage datasets have problems like erratic resolution, inconsistent picture data capturing equipment and extrinsic influences such as lighting and shadows. These have a significant impact on the criteria for the datasets used to train the models. Therefore, this study used a pavement damage dataset that was collected and produced by us. The initial image acquisition device is an integrated vehicle used for pavement detection, as shown in Figure 5. The parameters of the on-board camera are shown in Table 1. Combined with the actual acquisition needs, the shooting height was set between 40 and 80 cm to ensure the right size of damage in the images. Images were captured under normal lighting for several asphalt as well as concrete roads, and then images with high clarity and a balanced amount of damage were manually retained for the next step of processing.

### 3.2. Data Processing

#### YOLOv5-Based Classification Network

Since this study focuses on the two most common types of road damage, potholes and cracks, the original data images collected need to be extracted and classified, i.e., two types of images with pothole and crack damage were selected to build the dataset. In order to achieve efficient and high accuracy classification, we adopted the YOLOv5 network with better performance for image classification. Four versions of the YOLOv5 model exist: YOLOv5s, YOLOv5m, YOLOv5l and YOLOv5x. After testing the four models, the smallest and quickest model, YOLOv5s, was used in this study under the condition of guaranteed accuracy. The acquired images were normalized and scaled down to 640 × 640 size prior to model training to ensure that the YOLO model performs optimally for training. After standardization, the data were manually annotated using annotation according to different types of road damage, where the annotation file format was txt. For data preparation, a total of 1000 images of potholes and fractures were prepared. During training, the learning rate was 0.01 and the mini-batch number and momentum coefficient were set to 2 and 0.937.

For the classification model, the true and predicted classification permutations are as follows: True Positives (*TP*): the number of true positive classes predicted as positive classes; False Positives (*FP*): the number of true negative classes predicted as positive classes; False Negatives (*FN*): the number of true positive classes predicted as negative classes; and True Negatives (*TN*): the number of true negative classes predicted as negative classes. The following indicators can be defined based on the values of the above four categories.

Accuracy is calculated as:(7)$$Accuracy=\frac{TP+TN}{TP+FP+TN+FN}$$

Precision is calculated as:(8)$$Precision=\frac{TP}{TP+FP}$$

Recall is calculated as:(9)$$Recall=\frac{TP}{TP+FN}$$

When the number of classification targets is unbalanced, the *F*1 *score* is used as the numerical evaluation index, and the *F*1 *score* is calculated as in Formula (10):(10)$$F_{1}score=\frac{2\times Precision\times Recall}{Precision+Recall}$$

After the training of the model, in the testing phase, the IOU threshold was set to 0.5 and the confidence threshold was set to 0.4. The final results were calculated according to the above formula and shown in Table 2. The classification accuracy of cracks and potholes reached 85.10% and 92.47%, and the F1 scores were 0.8512 and 0.9259, respectively. Most of the images of cracks and potholes can be correctly selected.

The images of potholes and cracks filtered by the classification network are shown in Figure 6. Data annotation was performed on these images to construct the dataset required for training. In total, there are 310 potholes and 300 cracks in the training set, 104 potholes and 101 cracks in the validation set and 103 potholes and 100 cracks in the test set.

## 4. Road-TransTrack-Based Road Damage Detection and Tracking

### 4.1. Model Initialization

Computer vision algorithms based on deep learning require an abundance of labeled images as datasets. Similarly, the transformer relies on a large amount of data. Studies and tests have shown that as the size of the dataset increases, the CNN model is eventually surpassed by the transformer model in terms of detection performance [44]. Thus, in order to improve the performance of the model, migration learning can be used to improve the model detection performance before training with the prepared dataset. Transfer learning is a way of improving learning effectiveness by transferring the knowledge structure of a related domain to the target domain [45]. In this study, a model that has been trained on the Microsoft COCO dataset was chosen for the transformer-based detection network setup.

### 4.2. Hyperparameter Tuning

In the case of deep learning networks, the model parameters include common parameters and hyperparameters. The common parameters are the weight parameters of each network layer. Back propagation and ongoing training can be used to find the best public parameters. Unlike public parameters, the values of the hyperparameters, which are generally set artificially through experience, were set before the start of training. In general, to enhance learning performance and effectiveness, manually optimizing the hyperparameters and selecting an ideal set of hyperparameters is required for model training. The hyperparameters that have an important effect on the model performance primarily comprise the learning rate, the weight decay coefficient and the mini-batch size. In this experiment, the learning rate and weight decay coefficients were adjusted, and six combinations were trained; the outcomes are shown in Table 3.

As shown in Table 3, the model obtained the highest accuracy when the learning rate was 2 × 10^−4^ and the weight decay coefficient was 10^−4^. As shown in Figure 7, the loss of the model gradually decreases as the training proceeds. As shown in Figure 8, the model accuracy gradually increases as the training proceeds. After 68 epochs of training, the model reached a maximum accuracy of 91.59% and the loss curve became flat. This model was saved and the hyperparameters set during the training of this model were selected for the next step of the study.

### 4.3. Transformer-Based Detection and Tracking Network

Some traditional CNN-based networks have achieved good performance in pavement image damage detection. However, for the same damage present in consecutive frames of video, these networks often either fail to detect it or perform duplicate counts without achieving good detection results. To address the above issues, the improved detection tracking network with a transformer was trained and tested on the dataset. Since the data are static images, adjacent frames are simulated during training by randomly scaling and transforming the static images. The optimal combination of hyperparameters derived in the above model initialization was selected for model training. As shown in Figure 9 and Figure 10, as the epoch increases, the loss value decreases, the accuracy increases and the optimal model is saved.

The upgraded tracking network was put to the test using the test set. Table 4 demonstrates the tracking network results; for the detection of pavement damage, the average accuracy score was 95.09% and the average F1 score value was 0.9646. As shown in Figure 11, the PR curve is close to the upper right corner of the coordinate system. This implies that the trained network performs well for tracking pavement diseases.

During the whole tracking process, the frame sequence is first detected. If the detection network detects the presence of damage in frame F_t_, the frame image is fed to the tracking network and the number of damages is increased by 1. Next, defects are detected in the next frame based on the features in frame F_t+1_ and are tracked based on the features in frame F_t_. Finally, IOU (Intersection over Union) matching is performed between the tracked and detected frames to obtain the tracking results.

To visualize the effects of model training more intuitively, two videos of pavement damage were selected to test the trained network.

As shown in Figure 12, two crack damages appear sequentially in the first video. The figure shows the tracking process from the appearance of the first crack to the appearance of the second crack and the simultaneous presence of both cracks, with the serial numbers of the two cracks in the upper left corner of the detection box in that order. The detection and counting results are consistent with the results of manual identification in the field.

As shown in Figure 13, three pothole damages appear in sequence in the second video. The diagram shows the tracking process from the appearance to disappearance of the first damage, the appearance to disappearance of the second damage, the coexistence of the first two damages and the appearance of the third damage, with the serial numbers of the three damages in the upper left corner of the detection box in that order. The detection and counting results are the same as those of manual identification in the field.

Both the above illustrations and results show that the trained model has good results for the detection tracking and counting of pavement potholes and crack damage and can basically meet the actual detection requirements.

## 5. Results and Discussion

In this study, the trained models were tested on a test set. To validate the performance of the model, the improved algorithm was compared with the CNN-based algorithm using evaluation metrics such as accuracy, precision, recall and F1 score. The algorithms YOLOv3, Single Shot MultiBox Detector (SSD) and Faster RCNN, which are commonly used for pavement damage detection, were selected for comparison [46,47]. For detection algorithms, the PR curve is an intuitive comparison graph; the closer the curve is to the upper right, the better the performance of the algorithm. As shown in Table 5, the classical CNN-based network was used to test the pavement damage dataset and the results were obtained. As shown in Table 6, compared with the classical CNN network, the F1 score of the detection network optimized by a transformer is the best, at 96.64, and the accuracy is 95.10%, which are 12.49% and 2.74% higher than the optimal CNN model, respectively. As illustrated in Figure 14a, for the class of cracks, when compared to various CNN models, the PR curve of the transformer optimized detection network is closest to the upper right corner. As illustrated in Figure 14b, for the class of potholes, the red curve of our network encloses the curves of YOLOv3, SSD and Faster RCNN. These comparative results effectively demonstrate that the transformer has a superior performance over CNN-based networks for the classification and detection of pavement damage.

In order to show the comparison results more intuitively, the same frame was detected with our network and the traditional CNN network, respectively, and the results were compared. As shown in Figure 15, for crack images, (a) the set of detection images demonstrates that the four networks detect approximately the same effect when there is only one crack in the figure and (b) the group detection images show that when multiple cracks appear in the figure, our network shows a better detection effect, without missing or wrong detections, and counts are carried out. For potholes, (a) the set of detection images shows that each network can accurately detect the two potholes present in the figure when the pothole size feature is obvious, and also our network counts the potholes and (b) the group detection images show that our network detects a pothole, while all other networks produce false detections, i.e., parts of the ground that are similar in shape to potholes are detected as potholes.

The above comparative tests show that the proposed model has good performance in terms of detection accuracy and accuracy of damage statistics. However, the detection speed of the current model does not meet the requirement of real-time execution. From the establishment of the dataset to the subsequent part of model testing, the current study uses pavement images and videos taken on the ground, so the generalization degree of the model needs to be further investigated. For example, the collection of pavement damage images can be carried out using the UAS technique to enrich the dataset required for model training; the model can then be used for the detection of images and videos captured by the UAS for better and efficient assessment of pavement damage.

## 6. Conclusions

For pavement damage video inspections, the detection accuracy is not high, resulting in the problem of repeated counting of missed detections. The main contribution of this study is the proposed tracking counting network called Road-TransTrack. When damage first appears in a video, it is detected and tracked until the defect disappears and the number of damages increases by 1. The tracking and counting model is improved with a transformer and a self-attention mechanism to improve the accuracy of damage detection and counting in road videos. Compared to the classic CNN network, the F1 score of the transformer-optimized detection network is 96.64, with an average accuracy of 95.10%, which are 12.49% and 2.74% higher than the optimal CNN model, respectively. A comparison of actual frame image detections shows that compared to other classical CNN networks, the model does not have the phenomena of missing and wrong detections. Additionally, the detection results of two road videos show that the model can track and count potholes and cracks correctly. All the above results indicate that the model in this study possesses better performance in video detection and tracking of road damage. In the future, we will consider training and testing models for more types of road damage.

## Figures and Tables

**Figure 1 sensors-23-07395-f001:**
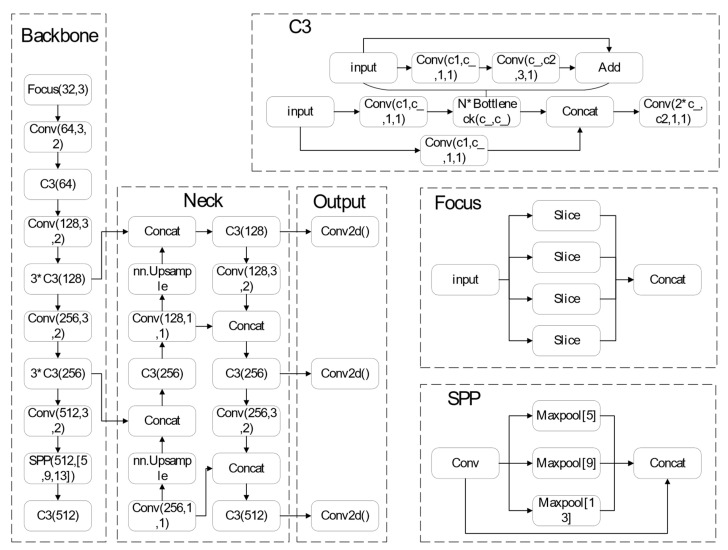
The detail of the network of YOLOv5s.

**Figure 2 sensors-23-07395-f002:**
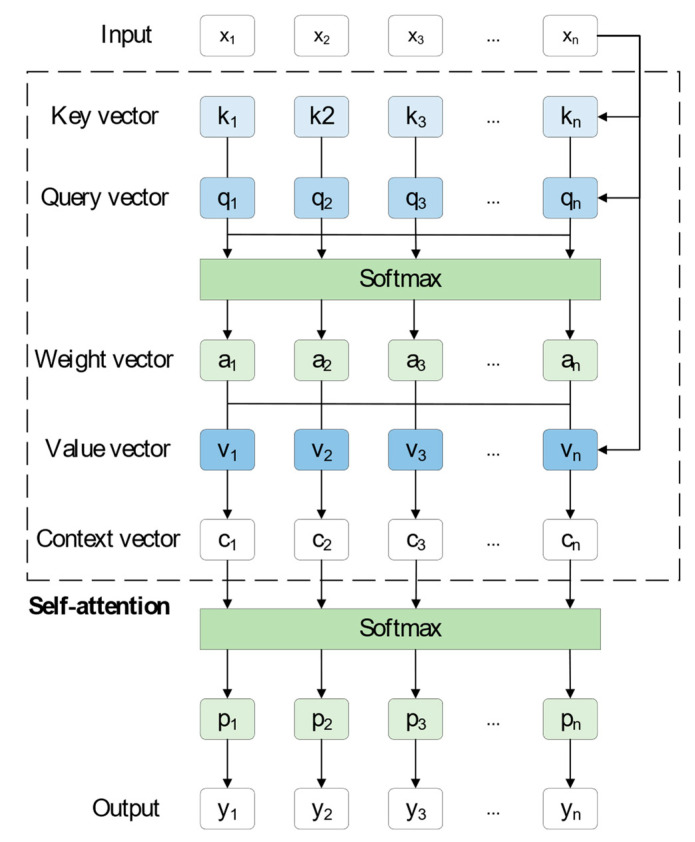
The self-attention mechanism’s structural diagram.

**Figure 3 sensors-23-07395-f003:**
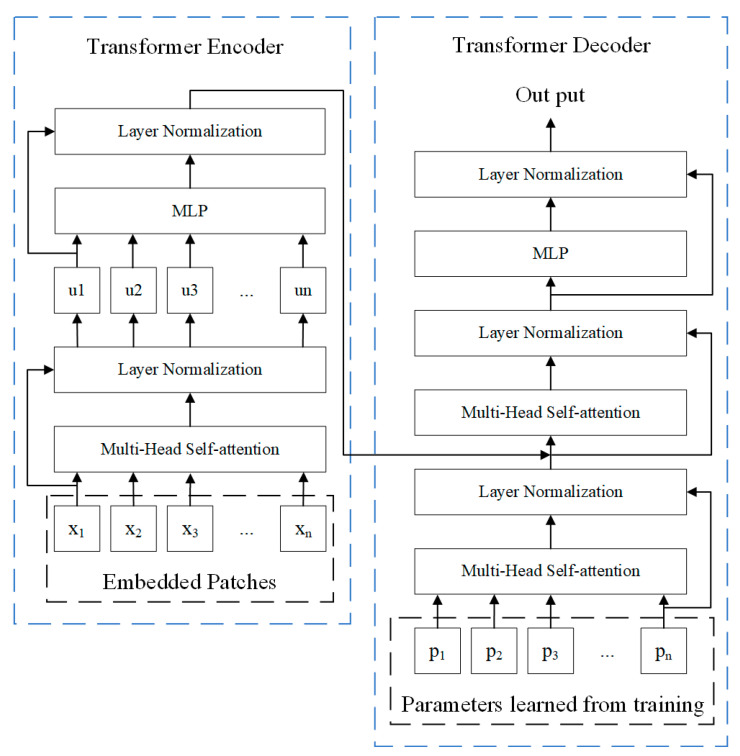
The structure diagram of the transformer.

**Figure 4 sensors-23-07395-f004:**
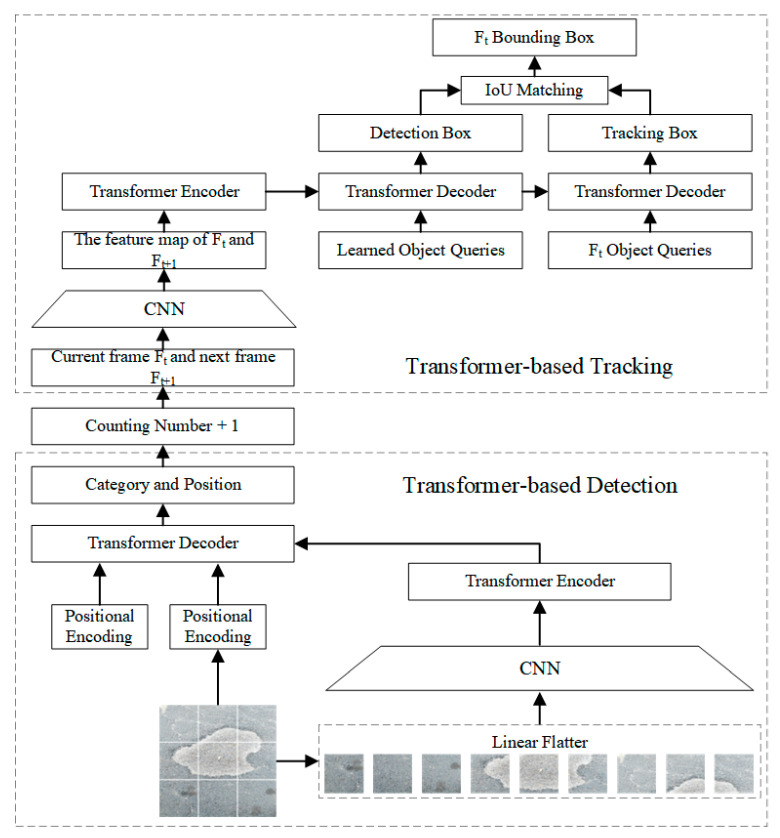
The detailed network of Road-TransTrack.

**Figure 5 sensors-23-07395-f005:**
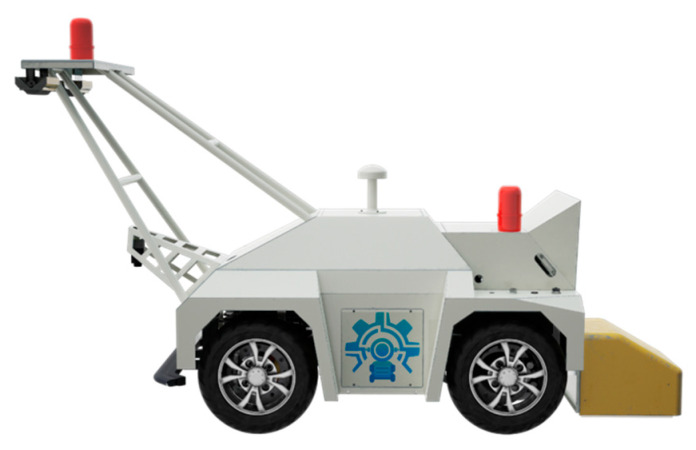
Integrated vehicle used for pavement detection.

**Figure 6 sensors-23-07395-f006:**
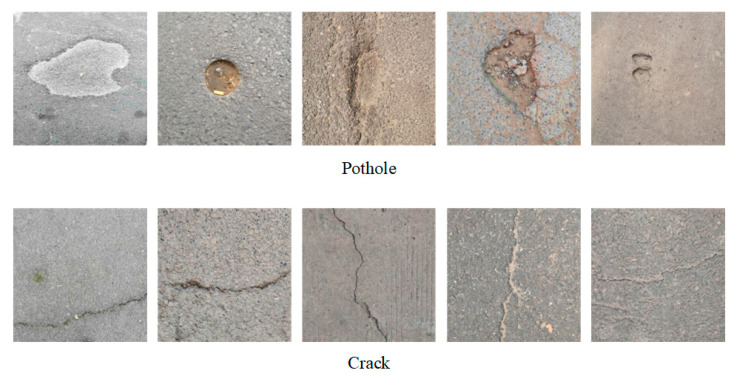
Damage images obtained from classification networks.

**Figure 7 sensors-23-07395-f007:**
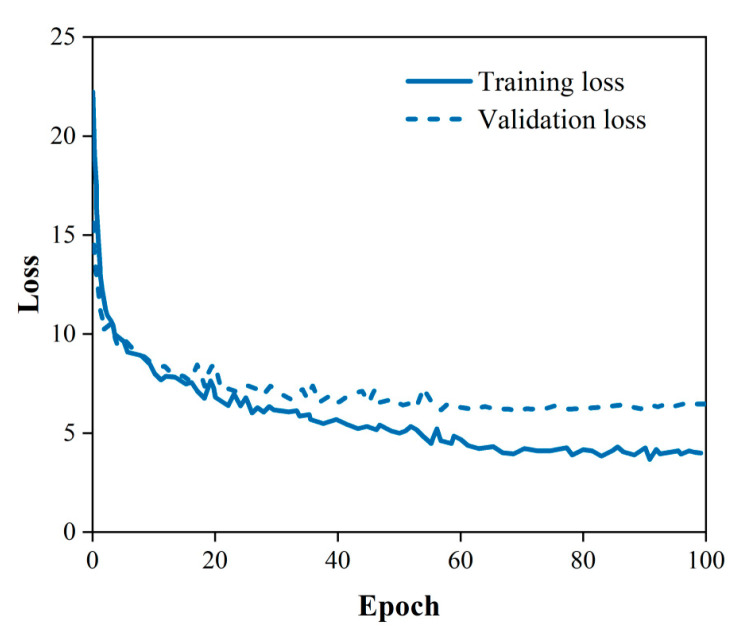
The decline curve of loss.

**Figure 8 sensors-23-07395-f008:**
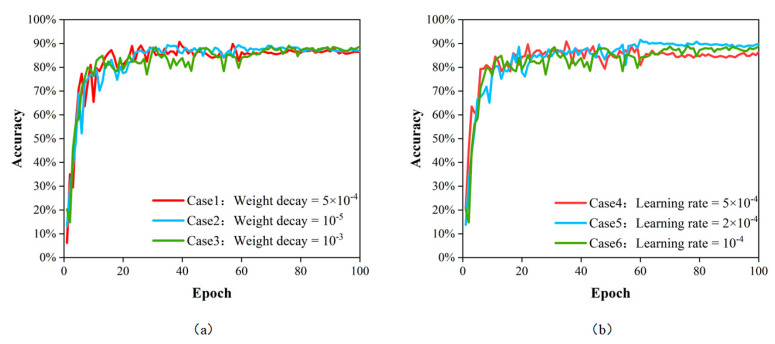
The upcurve of accuracy (**a**) with different weight decays; (**b**) with different learning rates.

**Figure 9 sensors-23-07395-f009:**
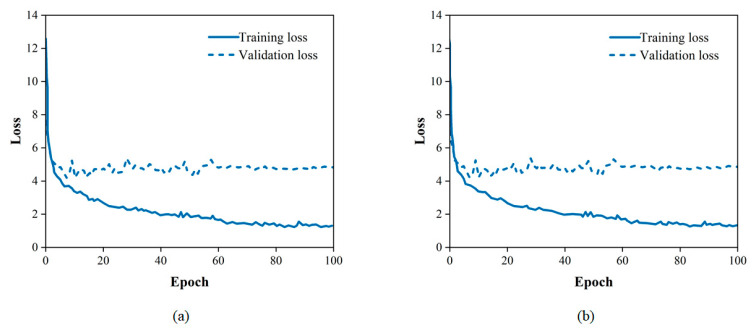
The decline curve of loss: (**a**) cracks; (**b**) potholes.

**Figure 10 sensors-23-07395-f010:**
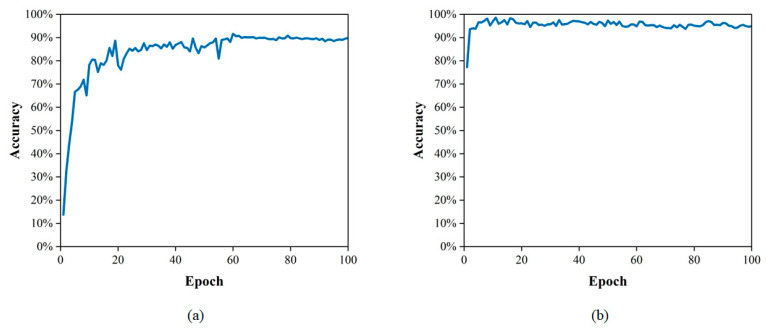
The upcurve of accuracy: (**a**) cracks; (**b**) potholes.

**Figure 11 sensors-23-07395-f011:**
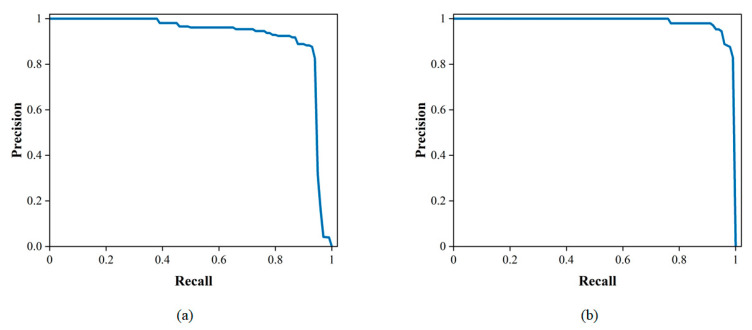
The PR curves of tracking network: (**a**) cracks; (**b**) potholes.

**Figure 12 sensors-23-07395-f012:**
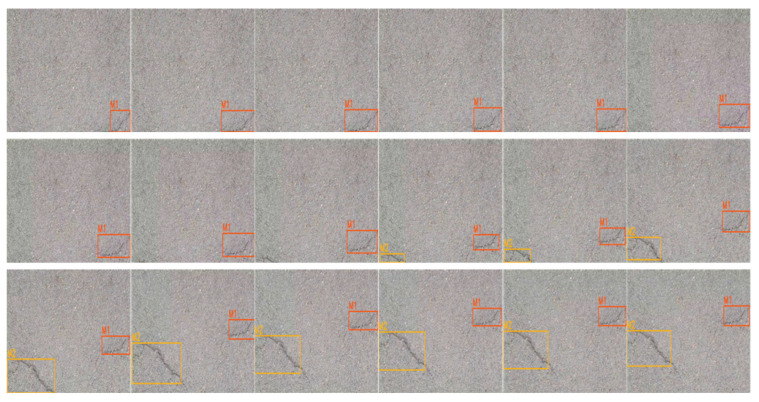
Tracking results for video 1.

**Figure 13 sensors-23-07395-f013:**
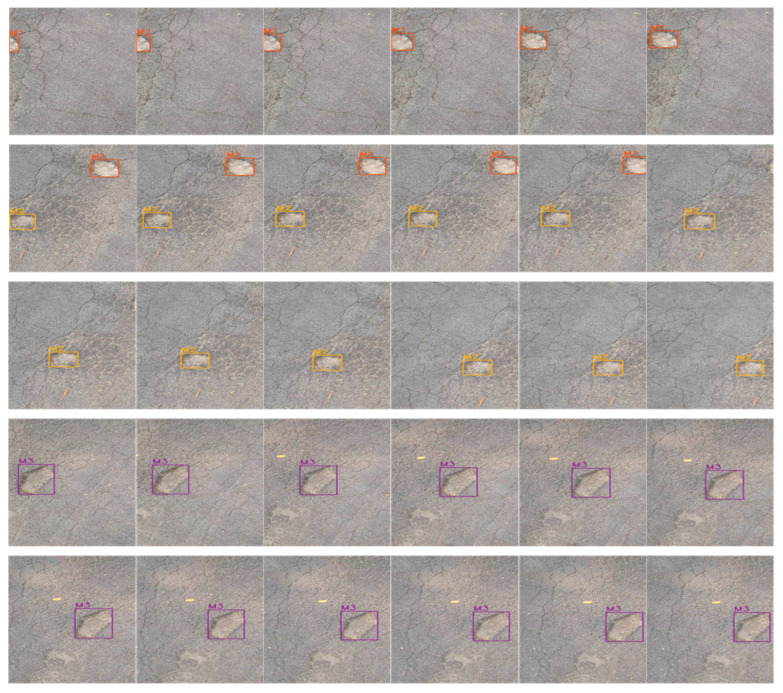
Tracking results for video 2.

**Figure 14 sensors-23-07395-f014:**
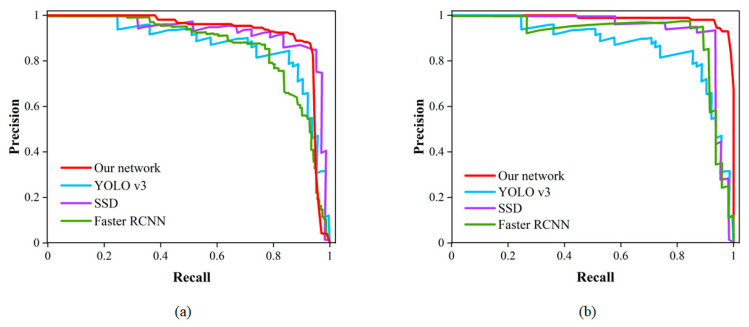
A comparison of PR curves: (**a**) cracks; (**b**) potholes.

**Figure 15 sensors-23-07395-f015:**
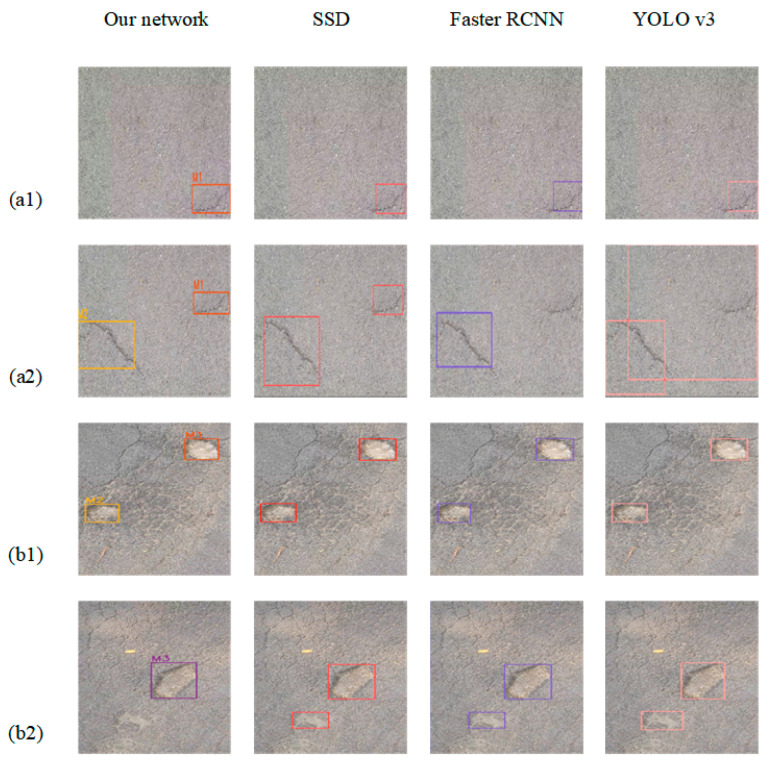
Comparison of different network detection results: (**a1**) comparison of individual crack detection results; (**a2**) comparison of multiple crack detection results; (**b1**) comparison of individual pothole detection results; (**b2**) comparison of multiple pothole detection results.

**Table 1 sensors-23-07395-t001:** Camera parameters.

Sensor Pictures	Equipment Parameters	HIKVISION U68
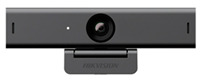	Highest resolution	4 K
	Highest resolution video output	3840 × 2160 30/25 FPS
	Maximum Field of View	83° × 91°
	Digital zoom	fourfold
	Autofocus	support
	TOF Sensing	support

**Table 2 sensors-23-07395-t002:** The output of the classification network based on YOLOV5.

Class	Accuracy	Precision	Recall	F1 Score
Crack	0.8510	0.8407	0.862	0.8512
Pothole	0.9247	0.9076	0.945	0.9259

**Table 3 sensors-23-07395-t003:** Hyperparameter tuning.

Case	Learning Rate	Weight Decay	Accuracy
1	10^−5^	5 × 10^−4^	90.73%
2	10^−5^	10^−5^	89.38%
3	10^−5^	10^−3^	89.75%
4	5 × 10^−5^	10^−4^	90.88%
5	2 × 10^−4^	10^−4^	91.59%
6	10^−4^	10^−4^	89.06%

**Table 4 sensors-23-07395-t004:** The results of the transformer-based tracking network.

Class	Accuracy	Precision	Recall	F1 Score
Crack	91.60%	91.6%	96.9%	0.9417
Pothole	98.59%	98.6%	98.9%	0.9874
Mean	95.095	95.1%	97.9%	0.9646

**Table 5 sensors-23-07395-t005:** The results of the classical CNN-based detection network.

Class	Accuracy	Precision	Recall	F1 Score
(YOLOv3)				
Crack	75.26%	75.06%	80.60%	77.73
Pothole	90.74%	89.56%	91.60%	90.57
(SSD)				
Crack	92.72%	91.49%	71.67%	80.37
Pothole	92.00%	80.39%	91.11%	85.41
(Faster RCNN)				
Crack	87.29%	35.37%	95.08%	51.55
Pothole	93.46%	75.00%	95.33%	83.95

**Table 6 sensors-23-07395-t006:** A comparison of different detection networks.

Network	Mean Precision	Mean Recall	Mean F1 Score	Mean Accuracy
Our Network	95.09%	97.90%	96.46	95.10%
YOLOv3	82.31%	86.10%	84.15	83.00%
SSD	85.94%	81.39%	82.98	92.36%
Faster RCNN	55.18%	95.21%	67.75	90.38%

## Data Availability

Data are contained within the article.

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
