# Peer review of "A Transformer-Optimized Deep Learning Network for Road Damage Detection and Tracking"

_sensors, 2023, doi:10.3390/s23177395_

Round 1

Reviewer 1 Report

The manuscript presents a deep learning network optimization for road damage detection.

Background theory and applied methodology are explained in detail.

Several acronyms are not defined on text.

Since it is a scientific manuscript, it should be written using the third person. Expressions like “we”, “our” must be replaced.

The construction of a dataset is valuable. However, the acquired images were obtained perpendicular to the ground plane (like aerial images from a drone) and do not have the perspective effect similar to the images obtained from a camera mounted in an inspection vehicle.  Have you think about that?

Line 131: The model parameters are not justified.

In the videos, authors show a video containing only cracks and another video denoting only potholes. Can the proposed approach deal with videos presenting both types of damage?

Could you refer in the manuscript if the proposed approach performs in real time or not?

There are no big issues with English language.

Reviewer 2 Report

A Transformer-optimized Deep Learning Network for Road 2 Damage Detection and Tracking

Review points

In this study, a tracking model named Road- 10 TransTrack based on transformer optimization is proposed to track and count the number of damages in road videos. The collected road damage images are classified into two categories, then the proposed tracking model is improved with a transformer and a self-attention mechanism. The title matches the content, and excellent work has been presented. I recommend the publication of the manuscript as it is.

Reviewer 3 Report

This research proposed a tracking model to track and count the number of damages on roadways. The model is named RoadTransTrack and it is developed based on transformer optimization. The classification model is developed based on YOLOv5, while the collected road damages are classified into two categories, including potholes and cracks. Research results showed that the proposed model can accurately detect potholes and cracks in the collected video. The reviewer believes that the current version of the manuscript is not yet ready for publication; the authors are encouraged to consider the following comments and suggestion and revise the manuscript accordingly.

1. The authors should streamline the Abstract section. Currently, it is very short and does not cover all of the required information. The Abstract section should focus on explaining why the research is needed, what the research is about, what the methodology is, and what the conclusion is. Do not include any unnecessary information but the required information must be provided.

2. The authors should consider reorganizing the manuscript to include the following sections: Introduction, Background, Methodology, Results and Discussion, and Conclusions. The Introduction section should focus on introducing the research objectives and stating the research questions that need to be answered, while the Background section should focus on reviewing of related literature and presenting the process of finding the research gap. The contents in the methods for YOLOv5-based classification network should be moved to the Introduction and Background section. The authors should also expand their literature review. For example, some of the key papers that using aerial photos for pavement surface evaluation were not reviewed and cited. For example, the authors should cite “Extracting pavement surface distress conditions based on high spatial resolution multispectral digital aerial photography”, and “Characterizing Pavement Surface Distress Conditions with Hyper-spatial Resolution Natural Color Aerial Photography”.

3. The authors should have a researcher that has a remote sensing background to proofread the manuscript. Many terms used in the manuscript are remote sensing related and the authors did not use them correctly. For example, in remote sensing there are four resolutions for an image, including spatial resolution, spectral resolution, temporal resolution, and radiometric resolution. The authors should use spatial resolution instead of resolution if that is what they want to indicate.

4. The authors should provide more detailed information about the camera (sensor) and the collected aerial videos (images). For example, what is the size of the sensor? What is the focal length of the camera? What is the spatial resolution of the collected imagery? Why are the images collected at 40-80 cm above the ground? What does the data collection system look like?  

5. The authors should discuss the potential of applying the proposed model to UAS collected imagery. This will significantly change the way of pavement condition assessment.

6. The authors should include a documentation for explaining their algorithms. Such a documentation will assist researchers in replicating the proposed method. The authors also need to go through the equations to make sure all elements in the equations are denoted.

7. Some figures and tables need to be improved. For example, Figure 1 and Figure 2 and Figure 3 are not readable and legible. If at all possible, the authors should use vector image for figure presentation.

Moderate editing of English language required.

Round 2

Reviewer 1 Report

Thank you for your efforts to address all my comments/suggestions.

The revised version has been much improved.

There are no big issues with English language.

Reviewer 3 Report

The authors have addressed all my comments.

N/A